# Survival Outcomes and Prognostic Factors in Rheumatoid Arthritis Patients Receiving Biologic or Targeted Synthetic Therapy: Real-World Data

**DOI:** 10.3390/antib14030054

**Published:** 2025-06-30

**Authors:** Zhaklin Apostolova, Tanya Shivacheva, Tsvetoslav Georgiev

**Affiliations:** 1First Department of Internal Medicine, Faculty of Medicine, Medical University–Varna, 9002 Varna, Bulgaria; zhaklin.apostolova@mu-varna.bg (Z.A.); tania.shivacheva@mu-varna.bg (T.S.); 2Rheumatology Department, St. Marina Hospital, 1, Hristo Smirnenski Blvd, 9010 Varna, Bulgaria

**Keywords:** rheumatoid arthritis, biological therapy, targeted antirheumatic agents, mortality, survival, prognosis, glucocorticoids

## Abstract

Objectives: The present study aimed to evaluate the long-term survival of patients with rheumatoid arthritis (RA) receiving biologic or targeted synthetic disease-modifying antirheumatic drugs (b/tsDMARDs) in a real-world setting, and to identify key prognostic factors influencing mortality within this cohort. Methods: This retrospective, observational cohort study analyzed 165 patients with confirmed RA who were on b/tsDMARD treatment for at least six months as of June 2017. Patient data, including demographics, disease duration, prior therapeutic regimens, and global functional status were extracted from medical records to collect data covering a seven-year follow-up period, extending from June 2017 to December 2024. Corticosteroid use was defined as continuous systemic intake during the RA activity analysis period. Survival outcomes were analyzed using Kaplan-Meier methods and multivariate Cox proportional hazards models to identify independent predictors of mortality. Results: Over a mean follow-up of 9.4 years, the mortality rate was 13.5 deaths per 1000 treatment-years, with an overall survival rate of 87.3%. Advanced functional disability and prolonged corticosteroid use were independently associated with higher mortality risk. In subgroup analyses, chronic kidney disease significantly increased mortality among patients on TNF inhibitors. In contrast, patients who remained on their initial anti-IL6 therapy had lower mortality, though this may reflect survivor bias. Conclusions: This study highlights the importance of long-term b/tsDMARD intervention in RA patients, with observed low mortality and high survival rates. Subgroup findings suggest the importance of comorbidity management in TNFi users and therapeutic stability in anti-IL6 users.

## 1. Introduction

Rheumatoid arthritis (RA) is a chronic autoimmune disease that significantly impacts quality of life and increases the risk of comorbidities and mortality [1]. The introduction of biologic and targeted synthetic disease-modifying antirheumatic drugs (b/tsDMARDs), along with modern therapeutic strategies, has substantially improved patient outcomes by reducing joint damage and disability [2]. Despite these advancements, RA patients still face a higher mortality rate compared to the general population, largely due to residual inflammation and associated comorbidities, particularly cardiovascular diseases [3,4,5,6,7].

Recent data suggest that new therapies may contribute to a lower mortality risk; however, findings remain inconsistent. Meanwhile, concerns persist regarding the long-term effects of immunosuppressive therapy, including susceptibility to severe infections and malignancies [8,9]. Although mortality in RA patients has declined over time, it remains elevated, highlighting the need for further improvements in disease management [10]. Comorbidity has been identified as a key modifiable factor in reducing mortality, with cardiovascular, respiratory, musculoskeletal, and digestive diseases accounting for a significant portion of RA-related deaths [11].

Several treatment strategies have been proposed to improve survival outcomes, including steroid-free therapy, early and targeted inflammation control, methotrexate use, and minimizing structural joint damage [12,13,14,15]. Timely initiation of b/tsDMARDs is particularly crucial in preserving joint function and reducing extra-articular complications [16]. Nevertheless, controlling inflammation alone is insufficient to close the survival gap. Integrated care—encompassing cardiovascular risk monitoring, lifestyle modifications, and addressing modifiable risk factors—should become a fundamental aspect of RA management.

Proinflammatory cytokines play a known role in RA pathogenesis and its extra-articular effects. However, the broader actions of anti–tumor necrosis factor (anti-TNF) and interleukin 6 (IL6) antibodies beyond joint control and destruction prevention are unclear. These antibodies help manage RA inflammation, leading to remission or low disease activity (LDA). Evidence suggests that anti-TNF antibodies also reduce mortality in RA patients [17]. Research on anti-TNF therapy and mortality in patients with RA has produced mixed results but generally suggests that anti-TNF therapy is not associated with an increased risk of all-cause mortality compared to traditional non-biological treatments. Some studies indicate that anti-TNF therapy may reduce mortality risk, particularly in women [10]. Patients treated with the TNFα antibody (adalimumab) had a lower number of deaths compared to expected age- and sex-matched individuals from the general population [18].

IL-6 has multiple metabolic and homeostatic effects extending beyond the site of inflammation. RA is linked to an increased risk of developing comorbid conditions, where IL-6 is also directly involved, including cardiovascular diseases, infections, osteoporosis, depression, neoplasia, anaemia, and higher mortality rates. IL-6 operates on various levels. Systemic inflammation and the acute phase response—measured by CRP levels and ESR—are associated with atherosclerosis. There are also established connections between IL-6 and cardiovascular disease [19]. Blockade of IL-6 is associated with numerous and diverse effects outside the joints. From reversing anaemia and depression to reducing atherosclerotic plaque and cardiovascular death [19].

How much of the reduction in mortality in RA patients treated with antibodies targeting TNF or IL-6 is due to suppression of active inflammation, and how much to mechanisms acting outside the known is still not entirely clear. There may be a potentiation of the effect of antibodies against TNF or IL-6 on inflammation with metabolic or other influences [20].

Survival data in RA from randomized clinical trials do not always reflect the complexity of real-world clinical practice. Studies show that patients in routine care settings often have more comorbidities, advanced disease stages, and varying adherence to therapy, which can lead to different outcomes compared to clinical trials.

Moreover, real-world data (RWD) is increasingly used to complement controlled clinical studies, providing valuable insights into treatment effectiveness in everyday practice. A national study in Lithuania compared mortality rates in patients with RA to the general population. The findings indicate that the risk of mortality in RA patients is 41% higher than in the general population [21]. A long-term 20-year study from the Australian Rheumatology Association Database (ARAD) revealed that RA-related mortality increases over time, with a standardized mortality ratio (SMR) of 1.49 after two decades. The leading causes of death include pneumonia and interstitial lung disease [22]. Additionally, a 7.5-year follow-up analysis of RA patients demonstrated a standardized mortality ratio (SMR) of 1.57, with malignancies being the most prevalent cause of death [6].

The primary aim of the study is to assess the long-term survival of patients with RA undergoing treatment with b/tsDMARDs using real-world data in accordance with the recommendations of the National Health Insurance Fund (NHIF). The secondary aim is to identify key mortality risk factors associated with baseline disease characteristics, concomitant medications, and comorbidities.

## 2. Materials and Methods

### 2.1. Study Design and Setting

This is a retrospective, observational, single-centre study conducted at the Rheumatology Clinic, St. Marina University Hospital, Varna, Bulgaria. The study evaluates the clinical characteristics and therapeutic efficacy in patients with RA, diagnosed according to the ACR criteria 1987 [23] and analyses their association with mortality outcomes.

### 2.2. Participants

#### 2.2.1. Sample Selection

Patients were selected sequentially from all RA cases monitored in 2017. Included patients had attended the diagnostic-consultative unit for continuation of outpatient biological DMARD therapy. All diagnoses were confirmed through complete medical records, including hospitalization summaries, outpatient consultations, and reviews by rheumatologists and other specialists.

#### 2.2.2. Inclusion and Exclusion Criteria

Inclusion criteria: Confirmed RA diagnosis according to the American College of Rheumatology (ACR) 1987 criteria [23].Ongoing biologic therapy for at least six months as of June 2017. Availability of comprehensive medical documentation, including:Demographic data (sex, age, BMI).Disease duration and prior therapeutic strategies.Radiographic staging of RA, assessed via conventional imaging, based on ACR classification, and extracted from medical records [24].Functional class, which describes the degree of functional limitation experienced by the patient, reflecting the overall impact of the disease on the individual’s ability to perform daily activities (I–III) [25].Compliance with National Health Insurance Fund (NHIF) criteria for b/tsDMARD Seropositive RA includes radiographic stage II or higher, an inadequate response to at least two conventional DMARDs (one being methotrexate at 20 mg/week), no malignancies, no severe hepatic or renal diseases, heart failure classification of NYHA class III or higher, and exclusion of patients in class IV [26].

The study exclusively includes patients who meet the predefined inclusion criteria, with all others being excluded.

#### 2.2.3. Patient Follow-Up

In 2017, an initial cohort of 209 patients diagnosed with RA was selected. Following the predefined eligibility criteria, 190 patients met all inclusion requirements. Between June 2017 and September 2018, disease activity assessments were evaluated at 6-month intervals (three visits for every patient) using the Simplified Disease Activity Index (SDAI) to monitor therapeutic response. By December 2024, a mortality analysis was performed on 165 patients. However, 25 patients lacked administrative follow-up data due to treatment discontinuation, referral to another healthcare facility, or loss to follow-up (Figure 1).

### 2.3. Assessment of Clinical Parameters

Of these 209 patients, 190 had complete medical records and met inclusion criteria, as well as having a full 18-month treatment period (2017–2018).

The SDAI is used to assess RA activity by combining clinical and laboratory parameters. It is calculated as the sum of the swollen joint count (SJC) and tender joint count (TJC), the patient-reported activity assessment (PGA), the physician-reported assessment (EGA), and serum C-reactive protein (CRP, mg/dL). The SDAI is used as a reliable tool in “treat-to-target” strategies, with clinical remission defined as values ≤ 3.3, low activity as values > 3.3 to ≤ 11, moderate activity as values > 11 to ≤ 26, and high activity as values > 26. In patients with established RA, as in our group, values ≤ 11 are considered acceptable. To assess the depth and duration of therapeutic outcome, all patients were evaluated at three consecutive visits over six months in the diagnostic and consulting room. With two or three consecutive SDAI values below 11, a 6-month or 12-month sustained LDA was accepted as the achieved therapeutic target [27].

### 2.4. Mortality Follow-Up

In December 2024, a mortality analysis was performed by searching the administrative database. From the initial sample, 165 patients were identified in the database—the remaining patients were likely those who discontinued therapy, transferred to another centre, or were otherwise lost to follow-up.

### 2.5. Definition of Variables

The primary outcome was mortality, assessed using the administrative database as of December 2024. The secondary outcome was the identification of key risk factors for mortality among patients with RA receiving b/tsDMARDs.

Exposures (Independent Variables) included demographic and clinical characteristics, such as age, sex, BMI, obesity, and smoking status. Disease duration and classification considered RA diagnosis before or after 2000, total disease duration, radiographic stage according to ACR classification, and functional class assessing mobility-related disability [25,28].

Treatment patterns included the time from diagnosis to initiation of b/tsDMARDs, the type of biologic therapy administered, assessed in 2017 and subsequently in December 2024, or until the time of death, baseline concomitant methotrexate treatment, Data regarding corticosteroid use were collected as a binary variable (yes/no), indicating whether patients were receiving systemic corticosteroid therapy at any point during the study observation period (June 2017 to September 2018), which coincided with the period of RA activity analysis. Prolonged corticosteroid use was defined as continuous intake throughout this activity assessment period. However, due to the retrospective nature of data collection, detailed information on cumulative doses, precise timing, or changes in corticosteroid use over the entire 9-year follow-up period was not available for analysis. This limitation should be considered when interpreting the effects of prolonged corticosteroid use on survival outcomes. The total duration of biologic therapy was evaluated until December 2024 or until the time of death. The presence of comorbidities, such as hypertension, diabetes, ischemic heart disease, chronic renal disease, pulmonary diseases, and other relevant conditions, was also assessed according to medical records.

### 2.6. Data Sources and Measurement Methods

Data Collection: Data were extracted from electronic and paper medical records at the Rheumatology Clinic, UMBAL “St. Marina”, Varna.

Mortality Assessment: Mortality data were obtained from the administrative database as of December 2024.

### 2.7. Selection Bias Evaluation

To minimize selection bias, patients were sequentially selected based on the order of visits from all monitored RA cases in 2017, reducing the risk of systematic bias.

To minimize immortal time bias, the start of follow-up (time zero) was defined individually for each patient based on the date of initiation of b/tsDMARD therapy. However, due to the retrospective nature of the study and limitations in the availability of historical treatment data, only patients who were alive and receiving b/tsDMARDs as of June 2017 could be included. Patients who initiated treatment before this date but discontinued therapy or died prior to mid-2017 were not captured. Consequently, the study population represents a subset of patients with stable long-term therapy and follow-up, and may not reflect the entire population initiating b/tsDMARD treatment. This limitation is addressed in the Discussion section.

Missing data, specifically for 25 patients (13% of the initial cohort) who lacked administrative follow-up data, were handled by listwise deletion in the final analysis. The assumption of Missing Completely at Random (MCAR) for these cases was assessed using Little’s MCAR test, which yielded a non-significant *p*-value (*p* = 1.000), supporting the MCAR assumption. Due to the absence of systematic bias, the subsequent analysis was conducted based on the group of 165 patients with complete information available as of 2024.

### 2.8. Statistical Methods

Descriptive Analysis: The main demographic and clinical characteristics of the patients were presented using descriptive statistics (means, standard deviations, and percentages). Comparisons were made using cross-tabulation and Chi-square, and for continuous variables using the Independent Student’s T test.

To assess the potential impact of missing data, multiple imputation using the Fully Conditional Specification method was applied (5 imputations, 10 iterations). The variable with missing values in 25 patients (13.2%) was examined to validate the assumption that the data were Missing Completely at Random (MCAR). As no significant differences were found in the results after imputation, all presented results are based on the original (non-imputed) dataset.

#### Survival Analysis (Kaplan-Meier)

Survival analysis was performed using the Kaplan-Meier method to estimate the time to all-cause mortality. The time variable was defined as the duration from the start of treatment until the date of death or the last follow-up (up to 31 December 2024), whichever occurred first. Patients who were alive at their last follow-up were considered censored. The restricted mean survival time (RMST), calculated as the area under the Kaplan-Meier curve up to the maximum observed follow-up time (19 years in this study), was used as a summary measure of survival. The RMST provides an estimate of the average event-free survival time within a period of treatment with b/tsDMARDs, accounting for all patients (both those who experienced an event and those who were censored). It differs from the simple average of treatment duration, as it provides a more comprehensive measure of expected survival within the study’s timeframe.

In addition, a Cox proportional hazards model was applied to assess the influence of the independent variables on the risk of mortality.

Statistical Significance: A *p*-value of <0.05 was considered statistically significant.

To enhance clarity and facilitate comprehension, the results were presented using both tabular and graphical methods.

## 3. Results

### 3.1. Baseline Clinical and Demographic Characteristics

The study included a total of 190 patients diagnosed with rheumatoid arthritis. Of these, 165 patients had complete data and were included in the primary analysis, while 25 patients had missing follow-up data. Table 1 summarizes the baseline clinical and demographic characteristics of all included patients, categorized by whether they were part of the analyzed cohort or had missing data.

As shown in Table 1, there were no statistically significant differences in the distribution of sex, BMI, smoking status, RA subgroups, or age between the analyzed patients and those with missing data (*p* > 0.05 for all comparisons). Similarly, baseline clinical characteristics such as duration of RA, type of biologic therapy, and use of methotrexate or methylprednisolone didn’t differ significantly between the two groups. Radiographic stage and functional class were also comparable. Regarding comorbidities, no significant differences were observed in the prevalence of chronic lung disease, ischemic heart disease, or chronic kidney disease (non-end-stage) between the analyzed and missing data groups. Finally, SDAI low disease activity, sustained at 6 and 12 months, also showed no significant differences between the two groups.

### 3.2. Patient Characteristics and Survival Outcomes

All patients enrolled in the study received b/tsDMARD therapy following the National Health Insurance Fund requirements for the treatment of seropositive active RA. Treatment was initiated between 2005 and January 2017, with follow-up extending until December 2024. The study included 165 RA patients (85.8% female) who were followed over a median treatment duration of 9.37 ± 2.44 years. A detailed characterization of the studied patients is presented in Table 2 and Table 3. These tables provide a comprehensive comparison of clinical characteristics, treatment history, and baseline comorbidities between RA patients who survived and those who did not, as of December 2024. No significant differences were observed between groups for demographic variables (age, sex, smoking, and BMI). A significantly higher proportion of non-survivors were assessed as functional class III—able to perform usual self-care activities but limited in occupational and recreational activities—compared to functional class II—able to perform usual self-care and occupational activities but limited in occupational activities (71.4% vs. 43.1%, *p* = 0.015). Furthermore, treatment-related differences were noted: non-survivors exhibited a significantly shorter duration of biologic therapy (6.66 vs. 9.77 years, *p* < 0.001) and higher rates of long-term methylprednisolone use (85.7% vs. 50.7%, *p* = 0.003). The results indicated that the initial b/tsDMARD regimen varied between groups, with non-survivors predominantly receiving anti-IL6 inhibitors (66.7% vs. 39.6%, *p* = 0.019) compared to anti-TNFi agents (33.3% vs. 60.4%, *p* = 0.019) at the baseline study. However, in the survival follow-up, no significant difference in therapeutic modality was found according to the type of b/tsDMARDs, and other treatment methods and comorbidities did not reach statistical significance (Table 2 and Table 3).

### 3.3. Baseline Patient Characteristics and Inter-Group Comparisons

Table 4 provides a detailed comparison of baseline demographic and clinical characteristics of the study cohort, stratified by their initial b/tsDMARD therapy: anti-IL6 inhibitors versus anti-TNF agents.

Notably, patients initiated on anti-IL6 inhibitors were significantly older when categorized (74.6% aged >55 years vs. 58.5% in the anti-TNF group, *p* = 0.031) compared to those on anti-TNF agents. Furthermore, the anti-IL6 group exhibited a significantly greater burden of disease severity and need for supportive care at baseline. Specifically, a significantly higher proportion of patients in the anti-IL6 group presented with more severe functional impairment, as evidenced by a higher percentage in Functional class III (56.3% vs. 39.4% in the anti-TNF group, *p* < 0.001). Correspondingly, the use of methylprednisolone was significantly more prevalent in the anti-IL6 inhibitor group (66.2% vs. 46.8% in the anti-TNF group, *p* = 0.013). A significant difference was also observed in X-ray stage (Stage III/IV: 81.7% in anti-IL6 vs. 55.3% in anti-TNF, *p* < 0.001), indicating potential variations in disease phenotype, with more severe radiographic damage in the anti-IL6 group.

No statistically significant differences were found between the groups for mean age (*p* = 0.061), mean BMI (*p* = 0.825), BMI >30 (*p* = 0.303), smoking status (*p* = 0.139), years from diagnosis to biologic treatment (*p* = 0.353), or total duration of RA (*p* = 0.691). Similarly, the prevalence of comorbidities such as arterial hypertension (*p* = 0.103), ischemic heart disease (*p* = 0.241), pulmonary diseases (*p* = 0.646), chronic kidney diseases (*p* = 0.582), and diabetes mellitus (*p* = 0.582) did not significantly differ between the two treatment groups. Additionally, there were no significant differences in the rates of sustained 6-month low disease activity (Sustained LDA, *p* = 0.929) or sustained 12-month low disease activity (Sustained 12-month LDA, *p* = 0.403) at baseline. These baseline distinctions, particularly in age, functional class, X-ray stage, and corticosteroid use, highlight important differences in the patient profiles that influenced initial treatment selection.

### 3.4. Therapeutic Transitions Between Initial and Last Known b/tsDMARD Regimens

A detailed c analysis was conducted to examine the dynamic flow of patients between their initial b/tsDMARD therapy (as of 2017) and their last known therapy received by December 2024 or at the time of death. This analysis revealed a statistically significant association between initial and last known therapy (Pearson Chi-Square = 12.521, df = 2, *p* = 0.002). Among patients initially treated with anti-TNF agents (N = 94), the majority (67.0%, n = 63) remained on anti-TNF therapy, while 25.5% (n = 24) transitioned to anti-IL6 inhibitors and 7.4% (n = 7) switched to JAK inhibitors. In contrast, for the cohort initially on anti-IL6 inhibitors (N = 71), a more pronounced pattern of switching was observed: 52.1% (n = 37) transitioned to anti-TNF therapy, while 47.9% (n = 34) continued on anti-IL6. No patients in the initial anti-IL6 group were recorded as transitioning to JAK inhibitors by the end of the follow-up period.

While the anti-IL6 inhibitor subgroup initially presented with more severe disease compared to the anti-TNF subgroup (evidenced by higher percentages of radiographic stage III/IV and functional class III, as presented in Table 3), levels of sustained therapeutic response (SDAI < 11 at 6 and 12 months) were similar between both groups. Nevertheless, a significantly smaller proportion of patients in the anti-IL6 group remained on their initial therapy compared to the anti-TNF group (47.9% versus 69%, *p* = 0.014), indicating a higher rate of treatment switches in the former.

These findings highlight the dynamic nature of treatment pathways in real-world RA management, where therapy modifications are common over long follow-up periods (Figure 2).

### 3.5. Kaplan–Meier Analysis of Treatment Duration and Survival Rate in RA Patients

The analysis of the imputed datasets revealed no significant differences compared to the original data. The estimated mean survival time and the proportion of censored cases remained consistent across imputations, confirming the robustness of the findings. Therefore, all presented results are based on the original (non-imputed) dataset.

Based on the Kaplan–Meier analysis for treatment duration (mean = 9.4 years, SD = 2.4), among 165 patients, there were 21 all-cause deaths (12.7%) and 144 survivors (87.3%). The Kaplan–Meier survival curve showed a high survival rate, with 87.3% (95% CI: 81.86–92.14%) of patients surviving throughout the entire follow-up period. The estimated restricted mean survival time (RMST) for treatment was 17.04 years (95% CI: 16.18–17.89), calculated as the area under the Kaplan-Meier curve. It is crucial to clarify that this model-based estimate of RMST extends beyond the observed mean treatment duration (9.4 years) and represents the average event-free survival within the maximum observed follow-up period (up to 19 years in this study). The substantially longer RMST reflects the large proportion of censored patients (87.3%) who remained alive and event-free for extended periods after the mean treatment completion. Interpretation of this estimate should acknowledge that it assumes the observed survival pattern continues throughout the entire period for which the curve is estimated. Considering the total follow-up duration of approximately 1551 treatment-years, the data correspond to approximately 13.5 death cases per 1000 treatment-years.

### 3.6. Multivariate Survival Analysis: Cox Proportional Hazards Model

In a multivariate Cox regression analysis evaluating risk factors for mortality in patients with RA receiving long-term treatment with b/tsDMARDs, baseline functional class, defined as class III, which can perform usual self-care activities but is limited in occupational and recreational activities, along with long-term corticosteroid use, were independently associated with mortality. Patients receiving methylprednisolone as a concomitant medication with b/tsDMARDs showed an approximately 5.5-fold increased risk of death (Adj. HR = 5.49; 95% CI: 1.61–18.68; *p* < 0.01). Conversely, any deterioration in functional class from II to III was linked to an approximately 2.75-fold increased risk of death (Adj HR = 2.75; 95% CI: 1.06–7.10; *p* = 0.037). The model included demographic factors (age, sex, BMI, obesity, smoking), clinical factors (diagnosed before and after 2000), and radiographic data, alongside concomitant methotrexate treatment, time from diagnosis to initiation of b/tsDMARDs, type of b/tsDMARDs (from 2017 to 2024), proportion of patients with a continuous 6- or 12-month LDA (2017–2018), and baseline comorbidities (including hypertension, ischemic heart disease, lung and kidney disease, and diabetes mellitus). The reported hazard ratio was adjusted for these factors, as they were not significant predictors of risk. All reported hazard ratios were adjusted for these potential confounders.

Collinearity analysis between the functional class of disability and methylprednisolone use was performed. The correlation coefficient (r = 0.003) suggests an extremely weak relationship between these two variables in the regression model. The predictors do not exhibit strong collinearity, meaning they largely act independently in explaining variance in the dependent variable.

### 3.7. Subgroup Analysis of Prognostic Factors by Initial Biologic/tsDMARD Therapy

To further investigate factors associated with all-cause mortality within specific treatment cohorts, Cox regression analyses were conducted for patients stratified by their initial b/tsDMARD therapy received at baseline.

In the anti-TNF inhibitor group (N = 94), with 7 events of mortality (7.4% of the subgroup), the multivariate Cox regression model identified two independent prognostic factors for mortality (Omnibus Tests of Model Coefficients: Chi-square = 11.067, df = 2, *p* = 0.004). Patients using methylprednisolone at baseline had a higher mortality risk (Adj HR = 9.568, 95% CI 1.100–83.211, *p* = 0.041). Those with chronic kidney disease at baseline also faced an increased mortality risk (Adj HR = 7.527, 95% CI 1.337–42.361, *p* = 0.022).

Other baseline characteristics, including age, BMI, smoking status, functional class, and other comorbidities, did not emerge as significant predictors of mortality in this specific subgroup analysis (*p*-values > 0.05 for these variables).

In the anti-IL6 inhibitor group (N = 71), with 14 deaths (19.7%), multivariate Cox regression identified the last b/tsDMARD therapy before death or before December 2024 as a significant mortality predictor (Chi-square = 4.330, df = 1, *p* = 0.037). Specifically, being on an anti-IL6 therapy at this final assessment time point was associated with a significantly reduced risk of mortality compared to being on an anti-TNF therapy (Adj HR = 0.257, 95% CI 0.072–0.924). No other baseline characteristics (including functional class, age, BMI, comorbidities, and others, as indicated by “Variables not in the Equation” all having *p*-values > 0.05) were identified as statistically significant independent prognostic factors for mortality in this subgroup.

## 4. Discussion

This real-world observational cohort study provides valuable insights into the long-term survival outcomes and prognostic factors among patients with RA treated with b/tsDMARDs. Over a mean follow-up of 9.37 years, an overall mortality rate of 13.5 deaths per 1000 treatment-years was observed, with an 87.3% survival rate. Our findings underscore that advanced functional disability and prolonged corticosteroid use remain independent and robust predictors of mortality in this patient population, consistent with previous research highlighting their detrimental impact on RA prognosis [3,16,29,30,31]. This emphasizes the critical importance of early and aggressive management strategies aimed at preserving the degree of limitation or impairment in a patient’s ability to perform routine tasks and minimizing corticosteroid dependency.

### 4.1. Comparison of Survival Outcomes with Published Literature

Our survival estimates are favorable compared to some previous data. For instance, a 2019 study reported a 5-year survival of 80% (95% CI: 78–81%) in RA patients [4]. While that study focused on a shorter follow-up and compared RA patients with general population controls, its findings underscore the persistent survival gap associated with RA. In contrast, our study, conducted in a population receiving modern biologic and targeted synthetic therapies, demonstrates relatively favorable long-term survival outcomes. Data from a Spanish study (2000–2004) found a mortality rate of 12.6 deaths per 1000 treatment years in patients receiving biologic therapy [32], a rate closely aligning with our observations. Similar trends have been described in a Lithuanian retrospective analysis and systematic review, noting that timely diagnosis and the frequent use of biologic therapy lead to significant improvements in survival despite RA patients having a higher overall mortality rate compared with the general population [21]. Our survival outcomes also align closely with data from the Swedish registry, which reported an incidence rate of all-cause mortality of 13.0 per 1000 person-years in RA patients initiating b/tsDMARD therapy between 2010 and 2020 [8]. These comparisons highlight the positive impact of modern RA management on patient longevity.

These comparisons highlight the positive impact of modern RA management on patient longevity. Nonetheless, it is important to acknowledge that despite therapeutic advances, RA patients continue to experience excess mortality compared to the general population. Although the present study did not aim to investigate specific causes of death, existing evidence consistently indicates that the leading causes of mortality in RA remain cardiovascular diseases, malignancies, and respiratory disorders—similar to those observed in the general population. Importantly, RA patients demonstrate increased mortality from all three causes.

While our focus on all-cause mortality provides a comprehensive overview of overall survival, it inherently limits the ability to discern specific causes of death (e.g., RA-related versus unrelated) and their potential differential associations with various therapies. Future research employing competing risks analysis would be valuable to clarify therapy-specific risks and provide a more granular understanding of mortality drivers in this patient population.

The Kaplan-Meier curve (Figure 3) was presented to visualize the overall long-term survival experience of the entire cohort. It is important to note that this curve was not stratified by key variables such as treatment type or functional class, nor were log-rank tests or landmark analyses performed for subgroup comparisons, as the primary objective was to assess overall cohort survival rather than comparative outcomes between specific subgroups.

### 4.2. Implications of Baseline Treatment Group Differences

The observed distinct baseline characteristics between the initial anti-IL6 and anti-TNF treatment groups warrant careful consideration, as detailed in Table 3. Patients initially treated with anti-IL6 inhibitors presented with a significantly more challenging clinical profile at baseline, including being significantly older (when categorized > 55 years), exhibiting more severe functional impairment (functional class III), and demonstrating greater radiographic damage (X-ray stage III/IV). A significantly higher proportion of patients in the anti-IL6 group were also utilizing methylprednisolone at baseline, suggesting a greater reliance on corticosteroids to manage their disease activity.

These findings strongly indicate that the initial choice of anti-IL6 inhibitors was not random but likely reflected a deliberate clinical decision to treat patients with more severe, established, or refractory RA. Given the established efficacy profile of anti-IL6 inhibitors in challenging cases of RA, particularly for those with inadequate responses to anti-TNF agents or systemic inflammatory manifestations, it is reasonable to assume that these agents were reserved for or preferred in patients presenting with a greater disease burden, older age, or more significant functional decline [33]. This underscores a critical aspect of real-world clinical practice: the selection of b/tsDMARDs is heavily influenced by the patient’s initial disease severity and complexity. Therefore, the observed initial imbalance in b/tsDMARD assignment is not necessarily indicative of a differential efficacy or safety profile impacting mortality in the long term, but rather reflects a strategic allocation of advanced therapies based on baseline patient characteristics. Despite these adjustments, residual confounding may persist, influencing comparisons between treatment subgroups.

### 4.3. Prognostic Factors in Specific Treatment Subgroups

To further explore the nuances of mortality risk within distinct initial therapy cohorts, we conducted subgroup analyses using Cox regression.

In the anti-TNF inhibitor group (N = 94), our analysis revealed that two specific baseline factors independently predicted mortality: Methylprednisolone use: Patients receiving methylprednisolone at baseline demonstrated a significantly increased risk of mortality (HR = 9.568). This finding aligns with the overall cohort analysis, reinforcing the persistent association between corticosteroid use and adverse outcomes, likely reflecting higher underlying disease activity or severity. Chronic kidney disease: The presence of chronic kidney disease at baseline was also identified as a significant independent predictor of mortality (HR = 7.527). This observation underscores the critical impact of severe comorbidities, particularly renal impairment, on survival outcomes in RA patients on anti-TNF therapy, indicating a vulnerability not fully mitigated by anti-TNF treatment [34]. Other baseline characteristics, including functional class, age, BMI, or smoking status, as well as comorbidities and CVR factors, did not reach statistical significance in this subgroup.

In contrast, in the anti-IL6 inhibitor group (N = 71), no traditional baseline demographic or clinical disease characteristics were found to be statistically significant predictors of mortality. However, multivariate Cox regression analysis identified the last known b/tsDMARD therapy received by December 2024 or at the date of death as a significant predictor (HR = 0.257, *p* = 0.037). Our analysis indicates that the observed Hazard Ratio (HR) pertains specifically to the last b/tsDMARD therapy received. It is crucial to acknowledge that the anti-IL6 subgroup initially presented with more severe disease, as evidenced by higher radiographic stage III/IV and functional class III, a detail elaborated in our Results section. Despite this baseline severity, sustained low disease activity (SDAI < 11) was comparably achieved and maintained between both the anti-IL6 and anti-TNF groups throughout the follow-up period.

Within the anti-IL6 cohort, we found that patients who continued their initial anti-IL6 therapy until the final assessment point (i.e., did not switch treatment) experienced a significantly reduced risk of mortality compared to those who switched to anti-TNF therapy. This suggests that the survival advantage seen in patients who remained on anti-IL6 therapy reflects a favorable and durable therapeutic response in a subgroup that successfully navigated their initial more severe disease. This finding highlights a form of survivor bias, where the observed benefit is linked to successful long-term retention of therapy due to sustained effectiveness, rather than an inherent, unconditioned effect of the initial drug choice or baseline health status. Notably, this complex dynamic appears more pronounced in the IL-6 inhibitor-treated subgroup compared to those on TNF inhibitors.

This intriguing result suggests a complex dynamic: patients who initially received anti-IL6 and remained on this therapy until the end of the study (i.e., did not require a drug switch) may represent a group with an improved prognosis compared to those who switched to anti-TNF therapy (but notably, not to JAK1i) by the study’s end [35]. This may indicate that specific patient profiles respond exceptionally well to IL-6 inhibition, leading to sustained efficacy and better long-term outcomes, or it may reflect successful disease management leading to stable therapy in this higher-risk group [36,37]. Further studies, including analyses of treatment sequences and reasons for switching therapies, would be essential to fully elucidate this observation

### 4.4. General Risk Factors and the Impact of Long-Term Targeted Therapy

Taken together, these subgroup analyses illuminate critical differences in prognostic profiles depending on the initial b/tsDMARD assigned. While methylprednisolone use and chronic kidney diseases were significant predictors of mortality in the anti-TNF group, no traditional baseline characteristics independently predicted mortality in the anti-IL6 group, except the final therapy received. This divergence in identified risk factors highlights the complex and potentially distinct pathways to adverse outcomes in different RA patient subsets. It suggests that while general prognostic indicators like corticosteroid use remain broadly relevant, the specific comorbidities or clinical features that drive mortality risk can vary considerably across patient groups that are stratified by their initial therapy choice. This observed heterogeneity underscores the need for a highly individualized approach to risk assessment and comorbidity management in RA, recognizing that the most impactful prognostic factors may differ based on the patient’s underlying disease severity, treatment history, and ongoing therapeutic regimen.

Our study’s findings regarding traditional cardiovascular and metabolic risk factors (age, sex, tobacco smoking, arterial hypertension, and diabetes mellitus) provide further insights. While these factors are typically associated with increased overall and cardiovascular mortality in the general population and RA, they did not emerge as significant predictors in our Cox analysis for this cohort. This contrasts with some earlier studies but aligns with a growing body of literature suggesting that the influence of classical risk factors may be attenuated under conditions of sustained, modern RA. Prolonged therapeutic intervention may modify the influence of classical risk factors, emphasizing the greater role of disease activity control and functional class in long-term survival. Several studies, however, indicate that their prognostic value may be attenuated under certain clinical conditions. For instance, in a large multi-biomarker study, baseline factors such as hypertension and diabetes were found to contribute to cardiovascular disease risk in RA patients, but their predictive strength was largely influenced by inflammatory burden and therapy effects [38]. Similarly, a narrative review of cardiovascular comorbidity in RA emphasizes that while factors like hyperlipidaemia and diabetes mellitus are established risk factors, their impact on outcomes can be mitigated by rigorous management strategies and early therapeutic interventions [39]. Moreover, recent reviews have highlighted that the high inflammatory burden in RA can modify the traditional risk relationships, such that effective anti-inflammatory treatment, evidenced by the overwhelming protective effect of sustained low disease activity achieved through biologic therapy, which may override the adverse effects of these comorbid conditions [37,40].

It is important to acknowledge that, due to the retrospective nature of our data collection, key comorbidities such as cardiovascular events, cancer, and serious infections were not quantitatively assessed or analyzed as time-dependent covariates. Furthermore, the binary classification of conditions like chronic kidney disease and diabetes mellitus lacked the granularity (e.g., staging for CKD) that more detailed prospective studies might provide, potentially limiting a comprehensive understanding of their dynamic impact on mortality outcomes.

Our data suggest that functional status and prolonged corticosteroid use hold significant prognostic value among patients with long-standing RA receiving targeted therapy for an average duration exceeding nine years. This implies that sustained therapeutic intervention can alter the impact of classical risk factors, highlighting that its primary aim extends beyond controlling disease activity to fundamentally preserving the patient’s functional status, specifically ensuring they maintain at least Class II functional ability to perform routine self-care and vocational activities—factors crucial for long-term survival. Although the HAQ-DI is a standard in academic research for evaluating functional ability, and its improvement is linked to a reduced mortality risk [41] our study employed the functional classification system. Given the established correlation between these instruments, we chose the latter due to its superior feasibility and direct relevance for real-world data, enabling a more straightforward assessment of the patient’s overall functional limitations in clinical practice [2,25,42]. Although pragmatic for real-world data collection and aligned with NHIF criteria, the use of a functional classification system (I–IV) to assess the degree of disability in patients with rheumatoid arthritis may inherently lack the sensitivity of more detailed and validated instruments such as the HAQ-DI.

In the present study, we utilized classes II and III, as these were the categories documented in the administrative databases of the analyzed patients. Patients in functional class I (without any limitations in physical capacity) and class IV (with severe functional impairment requiring assistance with daily activities) are not eligible for long-term outpatient treatment with innovative therapies and were therefore not included in the analysis. The critical role of functional ability is further underscored by its direct link to physical activity. Since functional disability in RA often limits physical activity, and physical activity itself is a known protective factor against mortality, maintaining functional capacity becomes paramount for patient survival [30] and b/tsDMARDs may enhance physical activity by reducing disease activity [31].

Regarding corticosteroid use, our study confirms its significant negative impact on mortality. Patients taking methylprednisolone had a multiple increased risk of death in both the overall study group and the subgroup with anti-TNF therapy, highlighting the harmful long-term effects of these agents, despite their initial benefits in controlling disease activity [3,43]. Their negative impact does not stop there, it is associated with a significant increase in mortality and consequently a shortened lifespan [3,5].

Notably, our findings did not confirm an anticipated direct relationship between baseline global functional class and structural damage (X-ray stage), assessed at the initiation of b/tsDMARDs, and disease activity. This discrepancy is likely attributable to the fact that disease activity was assessed later, during ongoing targeted therapy (specifically, sustained LDA between 2017–2018, approximately 3.5 years after baseline), which may have confounded a clearer association with the initial functional and structural status. Concurrently, this sustained LDA at distinct later time-points did not prove to be a significant determinant (protective or risk factor) of mortality in the subsequent 6 years of follow-up.

These findings contrast with some literature. For instance, in the BeSt study, analyses of disease activity measures during a 10-year trial period showed that ever achieving sustained remission (for a minimum of 1 year) was associated with a hazard ratio of 0.69 (95% CI 0.44 to 1.06) for mortality in the decade following the trial’s end. This suggests that long-term, well-controlled disease activity, as assessed in studies like BeSt with more frequent and prolonged activity measurements (median 35 out of 41 study visits), may indeed be a protective factor against mortality. Our differing results might reflect the challenge of maintaining LDA over extended periods, or suggest that other, more enduring factors exert a greater influence on long-term survival in our cohort [7,44].

### 4.5. Strengths and Limitations

Despite the inherent limitations of a retrospective, single-center cohort study, the present work possesses significant strengths that contribute valuable insights into the long-term management of RA. The study provides valuable real-world data for a large cohort of patients treated with b/tsDMARDs in accordance with the strict criteria of the National Health Insurance Fund (NHIF).

One of the primary strengths is the long-term period of therapy and, consequently, the long-term survival analysis, with a mean follow-up duration of 9.37 years, which allows for a reliable assessment of long-term survival and the identification of prognostic factors. The comprehensive dataset used, including functional status, radiographic damage, activity, and comorbidities, provides an in-depth characterization of the cohort.

However, the study has several limitations that warrant careful consideration. These include methodological constraints such as the relatively small sample size and the absence of a control group, which limit statistical power and generalizability. Potential biases, including selection bias, immortal time bias, and survivor bias, are inherent to the retrospective design. Furthermore, the scope of data collection limited our ability to conduct time-varying analyses, to quantitatively assess all key comorbidities as time-dependent covariates, or to capture unmeasured confounders like socioeconomic status, medication adherence, or frailty. The granularity of certain measures, such as the functional class system, also presents a limitation. The estimated restricted mean survival time of 17.04 years, while a model-based calculation, should be interpreted with caution as it extends beyond the observed average follow-up duration and assumes the observed survival pattern continues. These and other methodological considerations, including the interpretation of subgroup findings and the focus on all-cause mortality, are discussed in detail in the preceding sections of the Discussion.

Despite these limitations, the study contributes important insights into long-term survival in RA patients treated with advanced therapies, but findings should be validated in larger, multicenter cohorts.

## 5. Conclusions

This study provides data on the long-term survival and mortality of patients with rheumatoid arthritis treated with b/tsDMARDs. The results indicate that advanced functional class represents an important independent risk factor associated with an increased risk of death. Additionally, prolonged corticosteroid use appears to be linked to higher mortality. In patients receiving anti-TNF therapy, chronic kidney disease also emerges as a significant contributor to increased risk.

The need for treatment switching, particularly among patients treated with anti-IL6 inhibitors, may be associated with a higher risk of death.

The observed low mortality and relatively high survival rates emphasize the importance of sustained, consistent, and well-controlled treatment with b/tsDMARDs, as well as the need to preserve patients’ functional capacity. Furthermore, minimizing the long-term adverse effects of chronic corticosteroid therapy remains a key factor in improving survival outcomes in individuals with rheumatoid arthritis. These patients deserve optimal treatment and care to support their quality and length of life.

## Figures and Tables

**Figure 1 antibodies-14-00054-f001:**
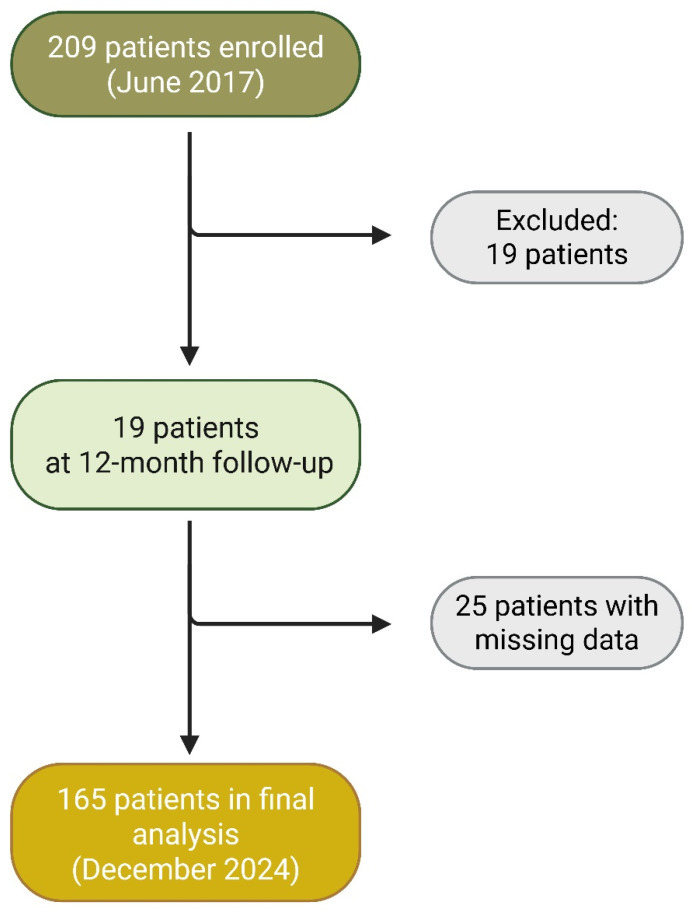
Study Flow Diagram: Cohort Selection and Follow-Up.

**Figure 2 antibodies-14-00054-f002:**
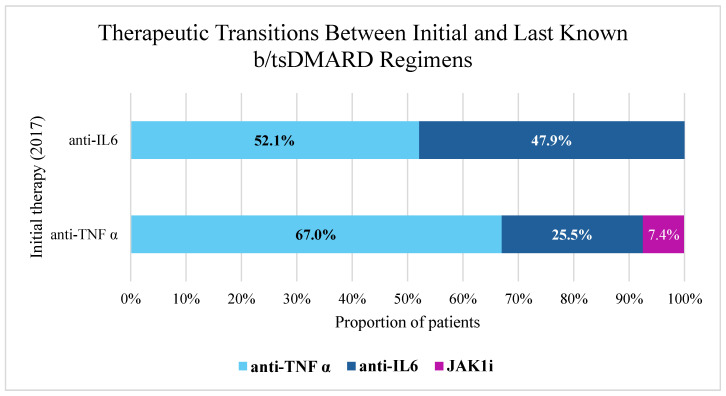
Therapeutic Transitions Between Initial and Last Known b/tsDMARD Regimens. Abbreviations: JAK1i—Janus Kinase 1 inhibitor; TNFi—Tumor Necrosis Factor inhibitor; IL6i—Interleukin-6 inhibitor.

**Figure 3 antibodies-14-00054-f003:**
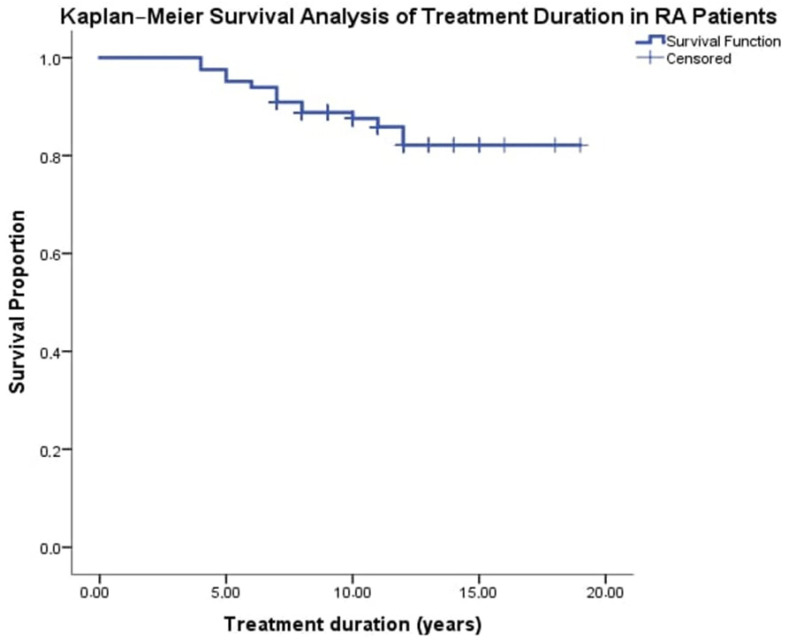
Kaplan–Meier Survival Analysis by b/tsDMARDs Treatment duration in RA Patients. Kaplan–Meier survival analysis graph for treatment duration in RA patients, the x-axis represents the treatment duration in years (ranging from 0 to 20 years), while the y-axis shows the cumulative survival probability (from 0.0 to 1.0). The blue survival curve illustrates how the probability of remaining event-free (i.e., surviving without the occurrence of death) decreases over time. Censored observations—patients for whom the event (death) was not observed during the follow-up period—are marked with blue crosses along the curve. This graph effectively provides a visual representation of the survival experience in the cohort, demonstrating both the overall survival trend and the timing of censoring, which is crucial when interpreting treatment duration outcomes.

**Table 1 antibodies-14-00054-t001:** Comparative Characteristics of Patients Included in the Study vs. Patients with Missing Data.

Parameter	All Included Patients (N = 190)	Analyzed Patients (N = 165)	Patients with Missing Data (N = 25)	*p*-Value *
Demographic Characteristics				
Sex, n (%)				0.213 †
Male	27 (14.2%)	26 (15.8%)	1 (4.0%)	
Female	163 (85.8%)	139 (84.2%)	24 (96.0%)	
BMI, n (%)				0.688 †
<30 kg/m^2^	146 (76.8%)	126 (76.4%)	20 (80.0%)	
≥30 kg/m^2^	44 (23.2%)	39 (23.6%)	5 (20.0%)	
Smoking Status, n (%)				0.447 †
No	132 (69.5%)	113 (68.5%)	19 (76.0%)	
Yes	58 (30.5%)	52 (31.5%)	6 (24.0%)	
RA Subgroups, n (%)				0.114 †
<2000 years	40 (21.1%)	38 (23.0%)	2 (8.0%)	
>2000 years	150 (78.9%)	127 (77.0%)	23 (92.0%)	
Age, years (mean ± SD)	64.38 ± 11.10	64.12 ± 11.40	66.12 ± 8.90	0.403
Clinical Characteristics at Baseline				
Duration of Treatment, years (mean ± SD)	N/A	9.36 ± 2.47	N/A	N/A
Duration of RA, years (mean ± SD)	N/A	18.07 ± 9.55	N/A	N/A
Type of Biologic Therapy, n (%)				0.282
anti-IL6	85 (44.7%)	71 (43.0%)	14 (56.0%)	
anti-TNF	105 (55.3%)	94 (57.0%)	11 (44.0%)	
Methotrexate Use, n (%)				1.000
No	73 (38.4%)	63 (38.2%)	10 (40.0%)	
Yes	117 (61.6%)	102 (61.8%)	15 (60.0%)	
Methylprednisolone Use, n (%)				0.132 †
No	81 (42.6%)	74 (44.8%)	7 (28.0%)	
Yes	109 (57.4%)	91 (55.2%)	18 (72.0%)	
Radiographic Stage, n (%)				0.507
II	65 (34.2%)	55 (33.3%)	10 (40.0%)	
III + IV	125 (65.8%)	110 (66.7%)	15 (60.0%)	
Functional Class, n (%)				0.668
II	103 (54.2%)	88 (53.3%)	15 (60.0%)	
III	87 (45.8%)	77 (46.7%)	10 (40.0%)	
Presence of Comorbidities, n (%)				
Arterial Hypertension, n (%)				0.387 †
No	78 (41.1%)	70 (42.4%)	8 (32.0%)	
Yes	112 (58.9%)	95 (57.6%)	17 (68.0%)	
Pulmonary Disease, n (%)				0.387 †
No	78 (41.1%)	70 (42.4%)	8 (32.0%)	
Yes	112 (58.9%)	95 (57.6%)	17 (68.0%)	
Ischemic Heart Disease, n (%)				0.321 †
No	167 (87.9%)	143 (86.7%)	24 (96.0%)	
Yes	23 (12.1%)	22 (13.3%)	1 (4.0%)	
Chronic Kidney Disease, n (%)				1.000 †
No	174 (91.6%)	151 (91.5%)	23 (92.0%)	
Yes	16 (8.4%)	14 (8.5%)	2 (8.0%)	
Diabetes Mellitus, n (%)				0.547 †
No	161 (84.7%)	141 (85.5%)	20 (80.0%)	
Yes	29 (15.3%)	24 (14.5%)	5 (20.0%)	
SDAI Low Disease Activity (2017–2018)				
Six-month LDA, n (%)				0.659 †
No	122 (64.2%)	107 (64.8%)	15 (60.0%)	
Yes	68 (35.8%)	58 (35.2%)	10 (40.0%)	
Twelve-month LDA, n (%)				0.493 †
No	128 (67.4%)	113 (68.5%)	15 (60.0%)	
Yes	62 (32.6%)	52 (31.5%)	(40.0%)	

Legend: *p*-values for continuous variables were calculated using an independent *t*-test, and for categorical variables using a Chi-square test, * *p*-value refers to comparison between analyzed group (N = 165) and missing patients (N = 25), † *p*-value calculated using Fisher’s Exact Test due to small expected frequencies or as recommended for 2 × 2 tables. Note: Disease activity data (SDAI) was evaluated for the period 2017–2018. Abbreviations: BMI: Body Mass Index; IL-6: Interleukin-6; LDA: Low Disease Activity; N/A: Not Applicable/Not Available; n: number (count); RA: Rheumatoid Arthritis; SD: Standard Deviation; SDAI: Simplified Disease Activity Index; TNF: Tumor Necrosis Factor; %: percentage.

**Table 2 antibodies-14-00054-t002:** Demographic and Baseline Characteristics of RA Patients by Survival Status.

Demographic and Baseline Characteristics	Survivors (N = 144)	Non-Survivors (N = 21)	*p*-Value
Age at the survival (years, mean ± SD)	63.8 ± 11.2	65.9 ± 12.6	NS
Female sex, n (%)	122 (87.8)	17 (81.0)	NS
Smokers	43 (29.9)	9 (42.9)	NS
BMI, mean ± SD	27.09 ± 4.91	27.42 ± 7.06	NS
BMI > 30, n (%)	33 (22.9)	6 (28.6)	NS
RA Diagnosis Year, n (%)			
Before 2000	31 (21.5)	7 (33.3)	NS
2000 or later	113 (78.5)	14 (66.7)	
RA Duration (years, mean ± SD)	18.03 ± 9.23	18.11 ± 9.77	NS
Radiographic stage III–IV, n (%)	93 (64.6)	17 (81.0)	NS
Functional class III, n (%)	62 (43.1)	15 (71.4)	0.015

Abbreviations: RA: Rheumatoid Arthritis, SD: Standard Deviation, BMI: Body Mass Index, NS: Not Statistically Significant, Statistical Analysis: Comparisons of continuous variables between survivors and non-survivors were performed using independent Student’s *t*-tests, while categorical variables were compared using chi-square tests.

**Table 3 antibodies-14-00054-t003:** Disease Management, Treatment History, and Baseline Comorbidities of RA Patients by Survival Status.

Parameter	Survivors (N = 144)	Non-Survivors (N = 21)	*p*-Value
Methotrexate Therapy, n (%)	91 (63.2)	11 (52.4)	NS
Methylprednisolone Use, n (%)	73 (50.7)	18 (85.7)	0.003
Mean Biologic Therapy Duration (years, mean ± SD)	9.77 ± 2.21	6.66 ± 2.28	<0.001
b/tsDMARDs at the beginning of the study			
Anti TNFi, n (%)	87 (60.4)	7 (33.3)	=0.019
Anti IL6i, n (%)	57 (39.6)	14 (66.7)	=0.019
b/tsDMARDs at Mortality Follow-Up			
Anti-TNFi, n (%)	84 (58.3)	16 (76.2)	NS
Anti-IL6i, n (%)	53 (36.8)	5 (23.8)	
JAK1i, n (%)	7 (4.9)	NA	
Level of Sustained SDAI LDA (assessed between 2017–2018)			
Twelve-month LDA, n (%)	49 (34.0)	3 (14.3)	NS
Six-month LDA, n (%)	26(18.1)	5 (23.8)	NS
One-moment LDA, n (%)	23 (16.0)	4 (19.0)	NS
No LDA, n (%)	46 (31.9)	9 (42.9)	NS
Comorbidities at Baseline			
Arterial Hypertension, n (%)	80 (55.6)	15 (71.4)	NS
Diabetes Mellitus, n (%)	21 (14.3)	3 (14.3)	NS
Ischemic Heart Disease, n (%)	18 (12.5)	4 (19.0)	NS
Chronic Kidney Disease, n (%)	10 (6.9)	4 (19.0)	NS
Pulmonary Disease, n (%)	10 (6.9)	NA	NA

Abbreviations: RA: Rheumatoid Arthritis, b/tsDMARDs: Biologic and Targeted Synthetic Disease-Modifying Antirheumatic Drugs, TNFi: Tumor Necrosis Factor Inhibitors, IL6i: Interleukin-6 Inhibitors, JAK1i: Janus Kinase 1 Inhibitors, SDAI: Simplified Disease Activity Index, LDA: Low Disease Activity, NS: Not Statistically Significant, NA: Not Applicable. Statistical Analysis: Comparisons between survivors and non-survivors were performed using independent Student’s *t*-tests for continuous variables and chi-square tests for categorical variables.

**Table 4 antibodies-14-00054-t004:** Baseline Demographic and Clinical Characteristics of Patients Stratified by Initial Biologic/Targeted Synthetic DMARD Therapy.

Characteristic	Anti-IL6 (N = 71) Mean ± SD or n (%)	Anti-TNF (N = 94) Mean ± SD or n (%)	*p*-Value
Age (years) (mean ± SD)	65.99 ± 10.09	62.71 ± 12.15	0.061
Age > 55 years, n (%)	53 (74.6%)	55 (58.5%)	0.031
Years from diagnosis to Biologic Treatment (mean ± SD)	9.45 ± 9.77	8.12 ± 8.57	0.353
Duration of RA (years) (mean ± SD)	18.41 ± 10.15	17.81 ± 9.12	0.691
BMI kg/m^2^ (mean ± SD)	26.97 ± 4.87	26.79 ± 5.67	0.825
BMI > 30, n (%)	14 (19.7%)	25 (26.6%)	0.303
Smoking (yes), n (%)	18 (25.4%)	34 (36.2%)	0.139
X-ray stage III/IV vs II n (%)	58 (81.7%)	52 (55.3%)	<0.001
Functional class III vs II, n (%)	40 (56.3%)	37 (39.4%)	<0.001
Methylprednisolone Use (yes), n (%)	47 (66.2%)	44 (46.8%)	0.013
Arterial hypertension (yes), n (%)	46 (64.8%)	49 (52.1%)	0.103
Ischemic Heart Disease (yes), n (%)	12 (16.9%)	10 (10.6%)	0.241
Pulmonary diseases (yes), n (%)	5 (7.0%)	5 (5.3%)	0.646
Chronic kidney diseases (yes), n (%)	7 (9.9%)	7 (7.4%)	0.582
Diabetes Mellitus (yes), n (%)	12 (16.9%)	12 (12.8%)	0.582
Sustained 6-month LDA (yes), n (%)	36 (50.7%)	47 (50.0%)	0.929
Sustained 12-month LDA (yes), n (%)	22 (31.0%)	35 (37.2%)	0.403

Abbreviations: RA: Rheumatoid Arthritis; BMI: Body Mass Index; LDA: Low Disease Activity; SDAI: Simplified Disease Activity Index. Statistical Analysis: Continuous variables are presented as Mean ± Standard Deviation (SD). Categorical variables are presented as a number of patients (n) and the percentage (%). Percentages are calculated within each treatment group (Anti-IL6 or Anti-TNF). *p*-values for continuous variables were calculated using Independent Samples *t*-tests. *p*-values for categorical variables were calculated using Pearson’s Chi-square tests.

## Data Availability

Data could be available by the corresponding upon reasonable request.

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
