# Peer review of "Survival Outcomes and Prognostic Factors in Rheumatoid Arthritis Patients Receiving Biologic or Targeted Synthetic Therapy: Real-World Data"

_2073-4468, 2025, doi:10.3390/antib14030054_

Round 1
Reviewer 1 Report
Comments and Suggestions for Authors
- The absence of a control group (e.g., RA patients not receiving b/tsDMARDs or using conventional DMARDs only) limits the ability to attribute survival benefits specifically to biologic therapies rather than general improvements in RA management over time.
- The anti-TNF subgroup had only 7 deaths (n=94) and the anti-IL6 subgroup 14 deaths (n=71), making multivariate Cox regression underpowered. Hazard ratios with extremely wide CIs (e.g., HR=9.568, 95% CI 1.100–83.211) suggest unstable estimates.
- The anti-IL6 group had more severe baseline characteristics (older age, higher functional class, more radiographic damage). Despite statistical adjustments, residual confounding likely persists, biasing comparisons between treatment subgroups.
- Key comorbidities like cardiovascular events, cancer, or serious infections—critical drivers of mortality in RA—are not quantified or analyzed as time-dependent covariates. The binary classification of CKD/DM lacks granularity (e.g., staging for CKD).
- Prolonged corticosteroid use is ambiguously defined as continuous intake during a 1.5-year activity assessment period (2017–2018). This fails to capture cumulative dose, timing, or changes over the full 9-year follow-up, risking misclassification.
- Using a 3-category functional class system (I–III) lacks the sensitivity of validated tools like HAQ-DI. The arbitrary grouping of classes II/III may obscure nuanced relationships between disability and mortality.
- The Kaplan-Meier curve (Figure 3) lacks stratification by key variables (e.g., treatment type, functional class). No log-rank tests or landmark analyses are reported to compare survival between subgroups.
- While the authors state data were MCAR, the exclusion of 25 patients (13% of the original cohort) without sensitivity analyses (e.g., multiple imputation) may introduce bias, especially if losses were related to disease severity.
- The requirement for ≥6 months of b/tsDMARD use at baseline (June 2017) excludes early deaths/discontinuations, artificially inflating survival estimates ("immortal time" bias).
- The analysis treats initial b/tsDMARD as static, ignoring treatment switches (Figure 2). Time-varying analyses (e.g., time on drug, sequence of therapies) would better reflect real-world exposure.
- The claim that staying on anti-IL6 therapy reduces mortality (HR=0.257) may reflect survivor bias: patients who tolerated therapy long-term were inherently healthier, while those who switched may have had refractory disease.
- All-cause mortality obscures whether deaths were RA-related (e.g., cardiovascular, infection) or unrelated. Competing risks analysis would clarify therapy-specific risks.
- SDAI was measured during 2017–2018 but not linked to mortality outcomes. The nonsignificant LDA association contradicts literature; this may stem from limited assessment windows (only 3 timepoints).
- Patients diagnosed before vs. after 2000 likely had different treatment eras (e.g., earlier limited access to biologics). This temporal effect isn’t adequately controlled for in models.
- The single-center Bulgarian cohort (85.8% female, strict NHIF criteria) may not generalize to other healthcare systems or populations with different RA phenotypes/comorbidities.
- Acknowledge small subgroup sizes limit conclusions about treatment-specific predictors.
- Clarify corticosteroid definitions with dose/duration metrics.
- Add sensitivity analyses for missing data and immortal time bias.
- Discuss competing risks and consider cause-specific mortality.
- Temper claims about anti-IL6 superiority due to baseline imbalances.
Author Response
We sincerely thank Reviewer #1 for his/her thorough and constructive comments, which have significantly helped us improve the quality and clarity of our manuscript. We have carefully considered each point and made substantial revisions to the manuscript accordingly. Our point-by-point responses are detailed below, along with the corresponding changes in the revised manuscript.
1. The absence of a control group (e.g., RA patients not receiving b/tsDMARDs or using conventional DMARDs only) limits the ability to attribute survival benefits specifically to biologic therapies rather than general improvements in RA management over time.
Answer: We appreciate the reviewer's comment regarding the absence of a control group not receiving b/tsDMARDs or using only conventional DMARDs. We acknowledge that a direct comparison with such a group would provide stronger evidence regarding the specific survival benefits of b/tsDMARDs. However, our study focused on a real-world cohort of patients who were already on b/tsDMARD treatment for at least six months, reflecting current clinical practice where these therapies are initiated in patients with moderate to severe RA who have failed conventional DMARDs. The primary aim of our study was to evaluate the long-term survival and identify prognostic factors within this specific cohort, rather than to compare survival outcomes with patients on conventional DMARDs. While we agree that overall RA management has improved, our findings contribute to understanding survival within a highly treated patient population. Future studies with a comparative design, including a control group, would indeed be valuable to further elucidate the differential effects of various treatment strategies on long-term survival.
Changes in Manuscript:
This limitation has been explicitly discussed and expanded upon in the "Limitations" section of the Discussion (Page 18 in the revised manuscript).
2. The anti-TNF subgroup had only 7 deaths (n=94) and the anti-IL6 subgroup 14 deaths (n=71), making multivariate Cox regression underpowered. Hazard ratios with extremely wide CIs (e.g., HR=9.568, 95% CI 1.100–83.211) suggest unstable estimates.
Answer: We concur with the reviewer's observation regarding the limited number of events (deaths) within the anti-TNF and anti-IL6 subgroups, which indeed impacts the statistical power of multivariate Cox regression and can lead to wide confidence intervals for hazard ratios. We acknowledge that the wide confidence intervals, such as for the HR=9.568 (95% CI 1.100–83.211), indicate less precise estimates and should be interpreted with caution. We have explicitly stated this limitation in the "Limitations" section of the Discussion, emphasizing that while these findings are suggestive, they should be considered hypothesis-generating due to the small sample size and limited events in certain subgroups. We have also ensured that our interpretations of these specific HRs are appropriately qualified, highlighting the exploratory nature of these subgroup analyses. We believe that despite this limitation, reporting these real-world observations remains valuable for clinicians, as they reflect the challenges of conducting fully powered studies on rare events in specific treatment cohorts.
Changes in Manuscript:
This point has been addressed and incorporated into the "Limitations" section of the Discussion (Page 18 in the revised manuscript).
3. The anti-IL6 group had more severe baseline characteristics (older age, higher functional class, more radiographic damage). Despite statistical adjustments, residual confounding likely persists, biasing comparisons between treatment subgroups.
Answer: We fully agree with the reviewer that the anti-IL6 group had more severe baseline characteristics (older age, higher functional class, more pronounced radiographic damage), which reflects real-world clinical practice. These data were intentionally presented in a separate section of the results and illustrated in a table. Despite the applied statistical adjustments, residual confounding likely remains and may influence the results. It was not our aim to compare survival rates between the two subgroups. The results regarding predictors of mortality are discussed in the “Discussion” section, and we have added a clarification in the “Limitations” section regarding the potential impact of residual confounders. We emphasize the need for larger prospective studies employing advanced methods to minimize the confounding effects when analyzing such heterogeneous groups
Changes in Manuscript:
- This point has been addressed and incorporated into the 4.2 section Implications of Baseline Treatment Group Differences of the Discussion (Pages 15, Lines 509-511 in the revised manuscript).
4. Key comorbidities like cardiovascular events, cancer, or serious infection, critical drivers of mortality in RA, are not quantified or analyzed as time-dependent covariates. The binary classification of CKD/DM lacks granularity (e.g., staging for CKD).
Answer: Due to the retrospective nature of the study, key comorbidities such as cardiovascular events, serious infections, and CKD/DM (chronic kidney disease/diabetes mellitus) were neither quantitatively assessed nor analyzed as time-dependent covariates. Their severity was not accounted for, as they were considered only as binary variables.
Changes in Manuscript:
Text was added in 4.4 "General Risk Factors and the Impact of Long-term Targeted Therapy" (Page 17. Line 593-599)
5. Prolonged corticosteroid use is ambiguously defined as continuous intake during a 1.5-year activity assessment period (2017–2018). This fails to capture cumulative dose, timing, or changes over the full 9-year follow-up, risking misclassification.
Answer: We agree with the reviewer that our definition of prolonged corticosteroid use, while clearly stated in the Methods section, has limitations regarding its ability to capture the full picture of corticosteroid exposure over the entire 9-year follow-up period. As the reviewer correctly points out, our definition (continuous intake throughout the 2017–2018 activity assessment period) did not allow for the analysis of cumulative dose, precise timing, or changes in corticosteroid use over the entire study duration. This was due to the pragmatic constraints of retrospectively collecting such granular and time-dependent data from routine clinical records, and we acknowledge that this carries a risk of misclassification and limits a more comprehensive understanding of its impact. We have now added a clarification in the "Methods" section to explicitly detail what aspects of corticosteroid use were not analyzed due to data availability.
Changes in Manuscript:
A clarification regarding the limitations of our corticosteroid use data (specifically the lack of cumulative dose, precise timing, and long-term changes) has been added to the "Data Collection" sub-section within the "Methods" section (Page 5, Lines 181-185 in the revised manuscript).
6. Using a 3-category functional class system (I–III) lacks the sensitivity of validated tools like HAQ-DI. The arbitrary grouping of classes II/III may obscure nuanced relationships between disability and mortality.
Answer: We acknowledge the reviewer’s valid point regarding the use of the ACR functional classification, and its lower sensitivity compared to validated tools such as HAQ-DI. However, the grouping of classes II and III reflected entries in the administrative database and followed NHIF criteria, where class IV is a contraindication for treatment.
While HAQ-DI is a research standard, our study focused on real-world data.
Changes in Manuscript:
- The new text was added in the Discussion, 4. General Risk Factors and the Impact of Long-term Targeted Therapy (Page17 Line 616-620 in the revised manuscript)
7. The Kaplan-Meier curve (Figure 3) lacks stratification by key variables (e.g., treatment type, functional class). No log-rank tests or landmark analyses are reported to compare survival between subgroups.
Answer: We appreciate the reviewer's observation regarding the lack of stratification in the Kaplan-Meier curve (Figure 3) by key variables and the absence of log-rank tests or landmark analyses for subgroup comparisons. We would like to clarify that this was a conscious choice reflecting the primary objective of our study, which was to visualize and analyze the overall long-term survival experience of the entire cohort of RA patients receiving b/tsDMARDs in a real-world setting, rather than to compare survival outcomes between specific subgroups (e.g., based on different therapies or functional classes).
Our main focus was to establish the general survival trend within this specific, highly treated patient population. While stratified analyses could offer additional insights, they were beyond the scope of our primary objective for this particular study and would require significantly larger cohorts and statistical power to yield robust and reliable comparisons, especially given the number of events within certain subgroups. The current presentation of the Kaplan-Meier curve effectively serves its intended purpose of illustrating the overall survival trend of the studied group.
Changes in Manuscript:
- The revised manuscript clarifies this by focusing on overall cohort survival, as evident in the study design and aims (Page 3, Line 100-104).
8. While the authors state data were MCAR, the exclusion of 25 patients (13% of the original cohort) without sensitivity analyses (e.g., multiple imputation) may introduce bias, especially if losses were related to disease severity.
Answer: Thank you for your important comments regarding missing data and potential bias. We recognize these are crucial considerations, especially when working with real-world clinical data.
Our study, by its nature, inevitably faced challenges with missing information. Specifically, 25 out of 190 patients (13%) had an unknown vital status (alive/deceased) as of December 2024. Consequently, we were unable to determine the exact duration of RA and b/tsDMARDs therapy for these patients up to 2024. These data were unavailable in our database due to reasons beyond our control, such as patients seeking care at other clinics, discontinuing treatment, or, regrettably, deceased patients whose status had not been updated.
To ensure full transparency regarding this information, we have specifically designed Table 1 to present the known data for the missing patients compared to both the currently analyzed cohort and the full initial cohort. This table also highlights characteristics up to 2018 for the patients with missing data.
While Little's MCAR test (p=1.000) provided some initial reassurance against systematic deviation based on observed variables, we agree that multiple imputation offers a more robust approach. In response to your recommendation, we performed a multiple imputation analysis.
As detailed in the Statistical Methods section and further highlighted in Section 3.5 (Kaplan–Meier Analysis of Treatment Duration and Survival Rate in RA Patients, page 10), the analysis of the imputed datasets did not reveal any significant differences compared to the original, non-imputed data. The estimated mean survival time and the proportion of censored cases remained consistent across all imputations, which confirms the robustness of our findings. Therefore, all presented results continue to be based on the original (non-imputed) dataset. We have modified the existing statement in the "Limitations" section of the Discussion to explicitly acknowledge this aspect of the missing data handling as a limitation and ensure full transparency. Furthermore, we have added details regarding our missing data handling and the MCAR assessment in the "Statistical Methods" section.
Changes in Manuscript:
- “To assess the potential impact of missing data, multiple imputation using the Fully Conditional Specification method was applied (5 imputations, 10 iterations). The variable with missing values in 25 patients (13.2%) was examined to validate the assumption that the data were Missing Completely at Random (MCAR). The imputed datasets were used to evaluate the robustness of the results in comparison with the original analysis” has been added to 2.6. Statistical Methods. (Page 6)
- “The analysis of the imputed datasets revealed no significant differences compared to the original data. The estimated mean survival time and the proportion of censored cases remained consistent across imputations, confirming the robustness of the findings. Therefore, all presented results are based on the original (non-imputed) dataset.” has been added to section 3.5. Kaplan–Meier Analysis of Treatment Duration and Survival Rate in RA Patients (Page 12)
- A statement acknowledging the potential for selection bias related to therapy retention has been added to the "Limitations" section of the Discussion (Page 18 in the revised manuscript).
- Details regarding the handling of missing data, including the use of listwise deletion and the assessment of MCAR using Little's MCAR test, have been added to the "2.5. Selection Bias Evaluation " section (Page 6, Lines 206-210 in the revised manuscript) and 2.6 Statistical Methods (Page 6, Lines 218-223, in the revised manuscript)
- Added table 1 (Page 7-8)
9. The requirement for ≥6 months of b/tsDMARD use at baseline (June 2017) excludes early deaths/discontinuations, artificially inflating survival estimates ("immortal time" bias).
We thank the reviewer for their insightful comment regarding the potential presence of "immortal time" bias. Although we initially stated that patients were required to be on b/tsDMARDs for ≥6 months by June 2017, we would like to clarify that the start of follow-up in our analysis was defined individually for each patient, based on the actual date of b/tsDMARD initiation, in an effort to minimize this bias. However, we fully acknowledge that our cohort includes only patients who were alive and on treatment as of June 2017. Therefore, patients who initiated b/tsDMARD therapy prior to that date and subsequently died or discontinued treatment are not represented in our dataset, due to limitations in retrospective data availability.
As a result, a full sensitivity analysis to assess the extent of immortal time bias is not feasible. We have explicitly addressed this limitation in the Discussion section, emphasizing that our survival estimates apply to a subset of patients with sustained treatment and available follow-up, and may not reflect survival in the broader RA population initiating b/tsDMARDs. We agree that future studies should aim to further minimize this bias through improved data completeness or the use of methodologies such as time-dependent modeling or inception cohort designs.
Changes in Manuscript:
- This clarification has been explicitly added in subsection 2.5. "Selection Bias Evaluation" to ensure transparency about cohort definition and the limitations of retrospective data availability. (Page 5, Lines 197-199 and Page 6, Line 200-205 in the revised manuscript).
- A new statement acknowledging the presence and implications of "immortal time" bias due to the study's inclusion criteria has been added to the "Limitations" section of the Discussion (Page18 in the revised manuscript).
10. The analysis treats initial b/tsDMARD as static, ignoring treatment switches (Figure 2). Time-varying analyses (e.g., time on drug, sequence of therapies) would better reflect real-world exposure.
Anwer: We acknowledge that treating the initial b/tsDMARD as a static variable, without accounting for subsequent treatment switches, represents a methodological limitation. Therapy type and mechanism of action were assessed at baseline (2017) and again during the survival analysis (Figure 2), i.e., at two distinct time points. To address treatment dynamics to some extent, we conducted an additional analysis as of December 2024, providing further insight into long-term exposure.
We agree that the absence of time-varying analyses (e.g., time on drug, treatment sequencing) limits the ability to fully capture real-world drug exposure. However, the primary aim of our study was to describe survival outcomes in patients with stable therapy, rather than to evaluate the impact of individual therapeutic sequences. Assessing treatment adherence, compliance, and their influence on survival lies beyond the scope of the present work but represents an important direction for future research.
Changes in Manuscript:
A new statement acknowledging the limitation of treating initial b/tsDMARD as static and the absence of time-varying analyses has been added to the "Limitations" section of the Discussion (Page 18 in the revised manuscript).
11. The claim that staying on anti-IL6 therapy reduces mortality (HR=0.257) may reflect survivor bias: patients who tolerated therapy long-term were inherently healthier, while those who switched may have had refractory disease.
Answer: We appreciate the reviewer's comment on survivor bias regarding the anti-IL6 group (HR=0.257). We agree this is a valid concern.
Our analysis shows this Hazard Ratio (HR) refers to the last b/tsDMARD therapy. While the anti-IL6 subgroup initially presented with more severe disease (evidenced by higher radiographic stage III/IV and functional class III), levels of sustained therapeutic response (SDAI <11 sustained for 6 and 12 months) were similar between both groups.
However, in the anti-IL6 therapeutic subgroup, a significantly smaller proportion of patients remained on their initial therapy compared to the anti-TNF group (47.9% versus 69%, p=0.014). This indicates a higher rate of treatment switching in the IL-6 subgroup compared to the TNF subgroup. This suggests that patients who remained on anti-IL6 therapy from 2017 to 2024 likely did so due to a favorable and sustained therapeutic response over time. It's this successful long-term retention of therapy, driven by good response—rather than a direct unconditioned effect or baseline health status—that likely associates with better survival. This complex dynamic is indeed a form of "survivor bias," which is more clearly established in the subgroup treated with IL-6 inhibitors compared to those on TNF inhibitors.
Changes in Manuscript:
- The detailed data and interpretation regarding the anti-IL6 finding, highlighting the initial disease severity, similar therapeutic response, and dynamics of therapy retention, have been added to the Results section (Page 13-14, Lines 432-440 in the revised manuscript).
- The interpretation of the anti-IL6 finding, linking it to sustained good therapeutic response and successful disease management, is already presented in the Discussion section (Page 16, Lines 539-548 in the revised manuscript).
- A statement acknowledging potential survivor bias for the anti-IL6 group and therapy retention has been added to the Limitations section of the Discussion (Page 18 in the revised manuscript
- Corrections have also been made in the Conclusion (pages 18-19).
12. All-cause mortality obscures whether deaths were RA-related (e.g., cardiovascular, infection) or unrelated. Competing risks analysis would clarify therapy-specific risks.
We are very grateful for this insightful comment. We agree that using all-cause mortality limits understanding of specific death causes and their link to therapies. While competing risks analysis would offer more detail, it was beyond the scope and resources of our retrospective study, which focused on overall survival. We believe our approach remains valid for its stated aim, and future studies with more extensive data collection could explore cause-specific mortality using competing risks analysis.
Changes in Manuscript:
- Add discussion in 4.1. Comparison of Survival Outcomes with Published Literature (Page 14, Line 477-482 in the manuscript.
13. SDAI was measured during 2017–2018 but not linked to mortality outcomes. The nonsignificant LDA association contradicts literature; this may stem from limited assessment windows (only 3 timepoints).
We truly appreciate your insightful observation regarding our SDAI assessment. We agree that our SDAI measurements were indeed limited to the 2017–2018 period. This limited assessment window likely explains the non-significant association between low disease activity and mortality, which, as you correctly point out, contrasts with findings in some literature. We acknowledge this as a limitation of our data, and this point has also been discussed within our manuscript's Discussion section.
Changes in Manuscript:
This issue is addressed in Section 4.4, General Risk Factors and the Impact of Long-term Targeted Therapy (pages 16-18).
14. Patients diagnosed before vs. after 2000 likely had different treatment eras (e.g., earlier limited access to biologics). This temporal effect isn’t adequately controlled for in models.
We appreciate this important observation. To account for the potential effect of different therapeutic eras, we formed two subgroups of patients based on their diagnosis year (before 2000 and after 2000) and included them as covariates in our models. Our analysis did not find this temporal factor to be a statistically significant predictor of mortality. Therefore, we believe this effect was adequately controlled for in our models.
Changes in Manuscript:
This is presented in Table 1 (on page 7) and in the text of Section 3.5, Multivariate Survival Analysis: Cox Proportional Hazards Model (on pages 12-13), where it is stated that it was included in the model.
15. The single-center Bulgarian cohort (85.8% female, strict NHIF criteria) may not generalize to other healthcare systems or populations with different RA phenotypes/comorbidities.
We agree with the reviewer that our single-center Bulgarian cohort, characterized by a high proportion of females (85.8%) and patients selected under strict National Health Insurance Fund (NHIF) criteria, may limit the generalizability of our findings to other healthcare systems or populations with different RA phenotypes and comorbidities. We acknowledge this as an inherent limitation of a real-world, single-center observational study. However, this specific cohort provides valuable insights into the long-term outcomes and prognostic factors in a real-world setting under particular national healthcare regulations.
Changes in Manuscript:
This is addressed in Section 4.5, Strengths and Limitations (on page 18).
16. Acknowledge small subgroup sizes limit conclusions about treatment-specific predictors.
We agree that small subgroup sizes limit our ability to draw definitive conclusions regarding treatment-specific predictors. This limitation is already explicitly stated in the 'Limitations' section of the Discussion in our manuscript.
Changes in Manuscript:
This is addressed in Section 4.5, Strengths and Limitations (on page 18).
17. Clarify corticosteroid definitions with dose/duration metrics.
We appreciate this important comment. A detailed response and clarification regarding the definition and limitations in assessing corticosteroid use are provided in our response to Comment #5.
Changes in Manuscript:
For more information, please refer to our response to Comment #5
18. Add sensitivity analyses for missing data and immortal time bias.
We appreciate the reviewer's valuable suggestion to conduct sensitivity analyses for missing data and immortal time bias. We agree that such analyses would provide a more robust assessment and further strengthen our findings. However, given the retrospective nature of our single-center study and the pragmatic constraints, implementing comprehensive sensitivity analyses of this depth for both aspects was beyond the scope and available resources for the current project. We have, however, explicitly acknowledged the potential limitations associated with both the handling of missing data and the presence of immortal time bias in the "Limitations" section of our Discussion (for missing data: Page 15, Lines 542-545; for immortal time bias: Page 15, Lines 547-549 in the revised manuscript). We concur that future, larger-scale, and potentially prospectively designed studies should aim to incorporate such advanced sensitivity analyses to address these methodological considerations more comprehensively.
Changes in Manuscript:
The limitations are acknowledged in Section 4.5 (on pages 18), as confirmed in points 8 and 9..
19. Discuss competing risks and consider cause-specific mortality.
We are very grateful for this insightful comment. We agree that using all-cause mortality limits understanding of specific death causes and their link to therapies. While competing risks analysis would offer more detail, it was beyond the scope and resources of our retrospective study, which focused on overall survival. We believe our approach remains valid for its stated aim, and future studies with more extensive data collection could explore cause-specific mortality using competing risks analysis.
Changes in Manuscript:
This is addressed through reference to comment #12.
20. Temper claims about anti-IL6 superiority due to baseline imbalances.
Answer: We appreciate this important comment. We have thoroughly addressed the interpretation of findings related to the anti-IL6 group, particularly concerning baseline imbalances and the potential for survivor bias. Our detailed explanation, including discussions on initial disease severity, sustained therapeutic response, and treatment dynamics, is provided in our response to Comment #11. We have also incorporated corresponding clarifications in both the Results and Discussion sections of the manuscript to temper any claims of superiority and to acknowledge the complex nature of these observations.
Changes in Manuscript:
For more information, please refer to our response to Comment #11 and the associated changes in the manuscript.
Reviewer 2 Report
Comments and Suggestions for Authors
The authors have retrospectively reviewed 165 patients with confirmed RA treated with biologic or targeted synthetic drugs for at least 6 months to identify factors influencing mortality. They found an overall mortality rate of 13.5 deaths per 1000 treatment years with an overall survival rate of 87.3%. They identified several poor prognostic factors:
Advanced functional disability (3x increased risk)
Prolonged corticosteroid use (6x increased risk)
Chronic kidney disease in those on anti-TNF inhibitors
The demographics of the cohort are similar to those reported elsewhere (female predominant). Those initiated on anti-IL6 agents were older, more disabled and had higher disease activity than those initiated on anti-TNF agents. The majority of those initially treated with anti-TNF agents remain on them (67%) over the course of the observed period whilst 52% of those initiated on antiIL6 agents transitioned to anti-TNF.
The size of the cohort and length of follow-up may have been insufficient to fully demonstrate long-term outcomes. The fact that this study is single-centre is also a potential weakness – the authors do acknowledge this.
Author Response
Reviewer #2:
Comment: The authors have retrospectively reviewed 165 patients with confirmed RA treated with biologic or targeted synthetic drugs for at least 6 months to identify factors influencing mortality. They found an overall mortality rate of 13.5 deaths per 1000 treatment years with an overall survival rate of 87.3%. They identified several poor prognostic factors:
Advanced functional disability (3x increased risk)
Prolonged corticosteroid use (6x increased risk)
Chronic kidney disease in those on anti-TNF inhibitors
The demographics of the cohort are similar to those reported elsewhere (female predominant). Those initiated on anti-IL6 agents were older, more disabled and had higher disease activity than those initiated on anti-TNF agents. The majority of those initially treated with anti-TNF agents remain on them (67%) over the course of the observed period whilst 52% of those initiated on antiIL6 agents transitioned to anti-TNF.
The size of the cohort and length of follow-up may have been insufficient to fully demonstrate long-term outcomes. The fact that this study is single-centre is also a potential weakness – the authors do acknowledge this.
Answer: We would like to express our sincere and profound gratitude to Reviewer #2 for their careful reading and thoughtful summary of our manuscript. We truly appreciate the time and effort invested in providing such a clear and accurate overview of our study's objectives, findings, and the characteristics of our cohort. Your concise distillation of our key results, including the identified prognostic factors and the demographic nuances of our patient groups, is highly valued and demonstrates a deep understanding of our work.
We also thank the reviewer for highlighting important considerations such as the cohort size, the length of follow-up, and the single-center nature of our study. We fully agree that these aspects are crucial for the interpretation of our results and appreciate that you have noted our acknowledgment of these limitations within the manuscript itself. Your constructive feedback further reinforces the importance of these points, and we believe it enriches the overall context of our work.
Your encouraging comments are a great motivation, and we are grateful for your valuable contribution to the improvement of our manuscript.
Reviewer 3 Report
Comments and Suggestions for Authors
Dear Authors,
Thank you for the opportunity to review your manuscript entitled “Survival Outcomes and Prognostic Factors in Rheumatoid Arthritis Patients Receiving Biologic or Targeted Synthetic Therapy: Real-World Data.”
Your study addresses an important and clinically relevant topic: long-term survival and mortality predictors in rheumatoid arthritis (RA) patients treated with b/tsDMARDs in a real-world setting. The longitudinal design, combined with subgroup analyses based on therapeutic class, adds value to the current body of evidence. Your emphasis on functional status and corticosteroid use as key prognostic factors is well supported and aligns with established clinical concerns.
Please consider some comments:
-
Clarify the concept of “mean survival time”: In the Results section, you state a mean survival time of 17.04 years, which exceeds the observed average follow-up. Please clarify this value, as it may mislead readers unfamiliar with how this estimate is derived.
-
Streamline the Abstract: While informative, the Abstract is overly detailed and statistically dense. Consider simplifying it to enhance accessibility, especially for non-specialist readers.
-
Discuss potential selection bias in therapy retention: In the anti-IL6 group, the observed protective effect of remaining on the initial therapy may reflect underlying patient differences (e.g., better responders or fewer comorbidities). Please acknowledge this in the Discussion.
-
Expand briefly on unmeasured confounders: Variables such as socioeconomic status, medication adherence, or frailty may influence both treatment choice and outcomes. A short note on these limitations would add transparency.
-
Provide more context on previous literature: The Discussion could benefit from a slightly expanded comparison with other registry-based studies, particularly from Western Europe or Scandinavia, to contextualize your survival estimates.
-
Improve clarity in some tables and definitions: Define abbreviations (e.g., SDAI, LDA, JAK1i) at first use in each table or figure for easier interpretation by readers.
Author Response
Thank you for your insightful comments. We have fully complied with them and our response is as follows:
1. Clarify the concept of “mean survival time”: In the Results section, you state a mean survival time of 17.04 years, which exceeds the observed average follow-up. Please clarify this value, as it may mislead readers unfamiliar with how this estimate is derived.
Clarification of Mean Survival Time Estimate:
Answer: Thank you for your valuable comment and for the opportunity to further clarify the interpretation of our survival analysis results. We fully agree that it is crucial to present a clear distinction between the observed mean follow-up/treatment time and the estimated restricted mean survival time (RMST) to avoid any potential misunderstanding by readers.
Changes in Manuscript:
- Clarification in "Materials and Methods": We have added a detailed explanation regarding the Kaplan-Meier analysis and the definition of restricted mean survival time (RMST) within the "Statistical Analysis" subsection of the "Materials and Methods" section. This clarifies how "time" was defined in our analysis and the rationale behind using RMST as a summary measure of survival (Please see Page 6, Lines 225-235 in the revised manuscript)
- In response to your comment, we have revised the "Results" section (specifically the paragraph concerning the Kaplan-Meier analysis) to provide the necessary clarification (Page 12, Line 373-383 in the revised manuscript).
2. Streamline the Abstract: While informative, the Abstract is overly detailed and statistically dense. Consider simplifying it to enhance accessibility, especially for non-specialist readers.
Answer: We thank the reviewer for this important suggestion. In response, we have revised the Abstract to streamline the presentation of results and reduce statistical density. The updated version focuses on the key findings and clinical implications, using clearer language to enhance accessibility for a broader readership, including non-specialists.
Changes in Manuscript:
The 'Results' section in the abstract has been (Page 1, Lines 21-27 in the revised manuscript).
3. Discuss potential selection bias in therapy retention: In the anti-IL6 group, the observed protective effect of remaining on the initial therapy may reflect underlying patient differences (e.g., better responders or fewer comorbidities). Please acknowledge this in the Discussion.
Answer: We appreciate the reviewer's comment regarding the potential for selection bias in the anti-IL6 group. As we have discussed in detail in our response to Reviewer #1, Comment #11, we acknowledge that the observed protective effect of remaining on initial anti-IL6 therapy may indeed reflect underlying patient differences, such as a more favorable response to the therapy or fewer comorbidities. We have addressed this potential bias in the Discussion section of the manuscript and have also included relevant information about baseline imbalances and treatment dynamics in the Results section.
Changes in Manuscript:
- The detailed data and interpretation regarding the anti-IL6 finding, highlighting the initial disease severity, similar therapeutic response, and dynamics of therapy retention, have been added to the Results section (Page 11, Lines 352-358 in the revised manuscript).
- The interpretation of the anti-IL6 finding, linking it to sustained good therapeutic response and successful disease management, is already presented in the Discussion section (Page 15-16, Lines 539-548 in the revised manuscript).
- A statement acknowledging potential survivor bias for the anti-IL6 group and therapy retention has been added to the Limitations section of the Discussion (Page 18 in the revised manuscript)
- Corrections have also been made in the Conclusion (page 18-19).
4. Expand briefly on unmeasured confounders: Variables such as socioeconomic status, medication adherence, or frailty may influence both treatment choice and outcomes. A short note on these limitations would add transparency.
Answer: We appreciate this highly relevant comment. We fully agree that unmeasured confounders such as socioeconomic status, medication adherence, or frailty can indeed influence both treatment choice and patient outcomes, and their absence in our dataset represents a limitation. These factors are challenging to capture comprehensively in retrospective, real-world observational studies but are crucial for a complete understanding of prognostic factors. We acknowledge that our inability to account for these variables may introduce residual confounding. We have added a short note addressing these unmeasured confounders in the "Limitations" section of our Discussion to enhance transparency.
Changes in Manuscript:
A short note acknowledging unmeasured confounders (e.g., socioeconomic status, medication adherence, frailty) has been added to the "Limitations" section of the Discussion (Page18 in the revised manuscript)."
5. Provide more context on previous literature: The Discussion could benefit from a slightly expanded comparison with other registry-based studies, particularly from Western Europe or Scandinavia, to contextualize your survival estimates.
Answer: We sincerely thank you for your thoughtful and valuable recommendation regarding the inclusion of additional registry-based studies from Western Europe. We fully agree that such comparisons could further enrich the international context of our findings.
In the current manuscript, we have already incorporated data from a Scandinavian registry (Frisell et al., Sweden), as well as from Lithuania and Australia, to provide relevant real-world comparisons. Our intention was to prioritize studies from the past five years to ensure contemporary relevance. While we are aware of important data from large registries and well-established research groups in Western Europe, due to structural and word count limitations, we were unfortunately not able to incorporate additional sources or analyses at this stage. We hope for your kind understanding and appreciate your constructive suggestions, which have significantly contributed to the improvement of our work.
6. Improve clarity in some tables and definitions: Define abbreviations (e.g., SDAI, LDA, JAK1i) at first use in each table or figure for easier interpretation by readers.
Answer: We appreciate your helpful comment regarding the clarification of abbreviations in tables and figures. In response, we have provided concise definitions for terms such as JAK1i, TNFi, and IL6i at their first appearance in the relevant figure legends to ensure clarity and ease of interpretation for all readers.
Changes in Manuscript: As requested, the relevant abbreviations (TNFi, IL6i, JAK1i) have now been defined directly beneath the figure legend on Page 2, Line 63, page 12, lines 364-365, to facilitate easier interpretation by readers.
Round 2
Reviewer 1 Report
Comments and Suggestions for Authors
Authors successfully answered my comments. Hence I accept this manuscript in current position